# HybridBNN: Joint Training of Deterministic and Stochastic Layers in Bayesian Neural Nets

**Amin Nejatbakhsh**                                     ANEJATBAKHSH@FLATIRONINSTITUTE.ORG
*Center for Computational Neuroscience, Flatiron Institute*

**Julien Boussard**                                                   JB4365@COLUMBIA.EDU
*Department of Statistics, Columbia University*

## Abstract

Bayesian Neural Nets are proposed as flexible models that can provide calibrated uncertainty estimates for out-of-distribution data. Due to the high dimensionality of BNN posteriors and the intractability of exact inference, numerous approximate inference techniques have been proposed. However, issues persist. Some approaches lack a proper Bayesian formulation while others result in inexpressive or uncalibrated posteriors, defeating the primary purpose of BNNs. Recently, subspace inference has been proposed to overcome these challenges by running the inference on a lower-dimensional subspace of network parameters. While achieving promising results, these methods are mathematically involved and therefore extending them to general architectures and problems is challenging. Here, we propose a new subspace inference method—called HybridBNN—that divides the network weights into deterministic and stochastic subsets before training. We develop an expectation-maximization algorithm for the joint inference of the posterior over the stochastic weights as well as the optimization of the deterministic ones. HybridBNN achieves competitive prediction and calibration performance on two regression and classification toy datasets and a benchmark dataset for in and out-of-domain distributions. The simplicity and flexibility of HybridBNN make it a favorable candidate for developing generic calibrated models.

## 1. Introduction

Developing calibrated predictive models that can flexibly fit any type of data has been a long-standing problem in statistical machine learning. While neural nets are known for providing overconfident predictions specifically for out-of-distribution data, several approaches have been proposed to alleviate their calibration issues primarily through Bayesian averaging (Minka, 2000). Instead of a single weight configuration that fits the data in a neural net, Bayesian averaging considers the space of all such weight configurations and weighs them according to their predictive performance. One popular way to perform this is through BNNs, which place a prior distribution over the weights of the neural net and infers the posterior distribution conditioned on the dataset.

$$\textbf{Prior } \boldsymbol{\theta} \sim \mathcal{N}(\mathbf{0}, \boldsymbol{I}), \quad \textbf{Likelihood } \boldsymbol{y}|\boldsymbol{x}, \boldsymbol{\theta} \sim p(\boldsymbol{y}|f_{\boldsymbol{\theta}}(\boldsymbol{x})), \quad \textbf{Posterior } p(\boldsymbol{\theta}|\{\boldsymbol{x}_i, \boldsymbol{y}_i\}_{i=1}^{N}) \quad (1)$$

The space of weights in BNN is high-dimensional; hence, exact inference is intractable. Therefore, approximate inference methods have been proposed. Initial proposals of inference algorithms for BNNs included Variational Inference (VI), Monte Carlo Dropout (MCD), and deep ensembles (Gal and Ghahramani, 2016; Blundell et al., 2015; Lakshminarayanan

et al., 2017). Unfortunately, all of these approximate inference methods are associated with well-established issues listed below. Despite achieving promising calibration performance, ensemble methods lack a proper Bayesian formulation. A recent paper shows that the in-distribution calibration power of an ensemble of $K$ networks with size $N$ is comparable to that of a single network of size $KN$ (Abe et al., 2022, 2023), therefore challenging the conventional understanding of ensembles as well-calibrated models. VI and MCD in BNNs are known to provide posteriors that are severely inexpressive (Foong et al., 2020). Furthermore, distance awareness, an important property of well-calibrated models, is absent in VI and MCD (Liu et al., 2020a). Laplace approximation and its variants alleviate some of the calibration issues associated with approximate inference in BNNs and provide a more cost-effective solution to posterior inference (Daxberger et al., 2021; Bergamin et al., 2024). Still, the inference is performed in the full network weight space which might be unnecessary.

Alternative to these approaches, Maddox et al. (2019) show that under strong assumptions, Stochastic Gradient Descent (SGD) can be cast as a sampling procedure. Osawa et al. (2019) build on this idea and show that this framework is flexible and hence can be combined with other deep learning tricks such as batch normalization, data augmentation, and learning rate scheduling. More recently, methods for performing inference in the subspaces of weights instead of full weight space are becoming more popular. Recent theoretical results suggest that partially stochastic BNNs have sufficient expressivity to capture the full posterior distribution of BNNs even in complex datasets (Sharma et al., 2023). Empirically, Izmailov et al. (2019) show that subspace inference in BNNs can indeed provide competitive performance and calibration. Daxberger et al. (2020) proposes another form of subspace inference called sub-network inference, which deviates from the original proposal by Izmailov et al. (2019) in both subspace construction and subspace inference. Most similar to our approach is Bajwa et al. (2020) in that we divide the space of weights into deterministic and stochastic and perform joint inference. However, their inference algorithm uses Bayes-by-Backprop, a form of variational inference that potentially leads to inexpressive posteriors (Blundell et al., 2015). Along similar lines, Kristiadi et al. (2020) show that `relu` BNN with stochastic nodes only in the last layer mitigates the overconfidence problem, but they use Laplace approximation which might be inexpressive in the general case.

Building on these works, we propose a new subspace inference method that divides the network weights into stochastic and deterministic ones, hence we call it HybridBNN. We treat the deterministic weights as the parameters of a statistical model and the stochastic weights as latent weights defining the space of functions mapping the inputs to the outputs. We develop an expectation maximization (EM) algorithm to jointly optimize the deterministic weights while inferring the posterior distribution over the stochastic weights. HybridBNN scales well to high dimensional inputs and provides competitive prediction and calibration performance on toy examples and a benchmark dataset.

## 2. Methods

Let $\boldsymbol{x}_i \in \mathbb{R}^d, \boldsymbol{y}_i \in \mathbb{R}^D, i = 1, \ldots, N$ denote the inputs and outputs of the HybridBNN. We represent the BNN weights by $\boldsymbol{\theta}$ and place a factorized Gaussian prior over them. Given a realization of $\boldsymbol{\theta}$ the BNN function is $f_{\boldsymbol{\theta}}(\boldsymbol{x}) \in \mathbb{R}^D$ and we aim to fit a predictive model

such that $\boldsymbol{y}|\boldsymbol{x}, \boldsymbol{\theta} \sim p(\boldsymbol{y}|f_{\boldsymbol{\theta}}(\boldsymbol{x}))$. We divide the weights of the neural network to $\boldsymbol{\theta} = \{\boldsymbol{\theta}_s, \boldsymbol{\theta}_d\}$ to represent stochastic and deterministic weights. Our key observation is that when the data is inherently low-dimensional, the space of functions representing the input-output relationship should also be low-dimensional, and the inference can be performed on a lower-dimensional space. We can treat the stochastic weights of the neural network as latent variables and the deterministic weights as the parameters in a probabilistic model, and reformulate our proposed method as a marginal likelihood optimization problem instead of posterior inference.

$$p_{\boldsymbol{\theta}_d}(\boldsymbol{x}_{1:N}, \boldsymbol{y}_{1:N}, \boldsymbol{\theta}_s) = p(\boldsymbol{\theta}_s) \prod_{i=1}^{N} p(\boldsymbol{x}_i) p(\boldsymbol{y}_i | f_{\boldsymbol{\theta}}(\boldsymbol{x}_i))$$

A natural choice for performing marginal likelihood optimization in this setting is to use the EM algorithm.

$$\max_{\boldsymbol{\theta}_d} \log p_{\boldsymbol{\theta}_d}(\boldsymbol{x}, \boldsymbol{y}) = \max_{\boldsymbol{\theta}_d} \log \int p_{\boldsymbol{\theta}_d}(\boldsymbol{x}, \boldsymbol{y}, \boldsymbol{\theta}_s) d\boldsymbol{\theta}_s = \max_{\boldsymbol{\theta}_d} \log \int p(\boldsymbol{x}) p(\boldsymbol{\theta}_s) p(\boldsymbol{y}|f_{\boldsymbol{\theta}}(\boldsymbol{x})) d\boldsymbol{\theta}_s$$

Below we elaborate on the **E** and **M** steps of the algorithm.

$$\underbrace{\mathcal{Q}(\boldsymbol{\theta}_d|\hat{\boldsymbol{\theta}}_d) = \mathbb{E}_{p(\boldsymbol{\theta}_s|\boldsymbol{x}, \boldsymbol{y}, \hat{\boldsymbol{\theta}}_d)}\left[ \log p(\boldsymbol{x}) p(\boldsymbol{\theta}_s) p(\boldsymbol{y}|f_{\boldsymbol{\theta}}(\boldsymbol{x})) d\boldsymbol{\theta}_s \right]}_{\textbf{E-Step}}, \quad \underbrace{\hat{\boldsymbol{\theta}}_d = \arg\max_{\boldsymbol{\theta}_d} \mathcal{Q}(\boldsymbol{\theta}_d|\hat{\boldsymbol{\theta}}_d)}_{\textbf{M-step}} \qquad (2)$$

For the **E** step, we use a Monte Carlo estimation of the integral shown below.

$$\mathcal{Q}(\boldsymbol{\theta}_d|\hat{\boldsymbol{\theta}}_d) \approx \frac{1}{M} \sum_{m=1}^{M} \left[ \log p(\boldsymbol{x}) p(\boldsymbol{\theta}_s^{(m)}) p(\boldsymbol{y}|f_{\boldsymbol{\theta}_s^{(m)}, \boldsymbol{\theta}_d}(\boldsymbol{x})) \right] \qquad (3)$$

where $M$ is the number of posterior samples. For the **M** step, we approximate the expected log joint using multiple samples drawn from the current estimate of the posterior. This approximate expectation is a function of the deterministic parameters, and we maximize it in the **M** step to update the deterministic parameters. Samples from the posterior are given by running MCMC in the low-dimensional stochastic weight space. We can compute the log of joint for each posterior sample as a function of deterministic parameters, simply by running a forward evaluation of the neural network and keeping track of its gradient wrt $\boldsymbol{\theta}_d$. The **M** step is performed simply by backpropagating the gradients of the following probabilistic loss wrt $\boldsymbol{\theta}_d$ and updating it.

$$\hat{\boldsymbol{\theta}}_d = \arg\max_{\boldsymbol{\theta}_d} \sum_{m,j=1}^{M,N} \log p(\boldsymbol{y}_j | f_{\boldsymbol{\theta}_s^{(m)}, \boldsymbol{\theta}_d}(\boldsymbol{x}_j))$$

In our experiments, we use `Adam` optimizer to perform this step. These steps can be repeated until convergence.

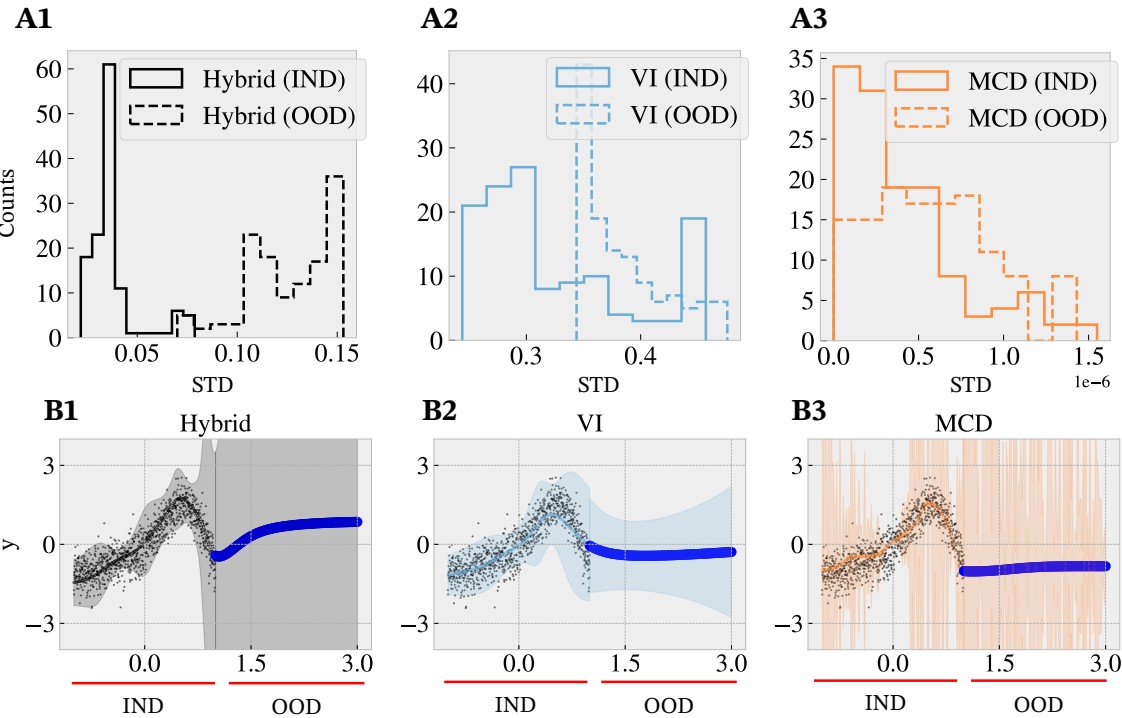

Figure 1: **Results on the toy regression model.** Data used for training is shown in the black boxes in panels (B1-B3) and model predictions are shown in colors (legends are consistent across two rows). All models perform well on mean prediction. (A1-A3) Standard deviation of the model predictions for in-distribution (IND) and out-of-distribution (OOD) data. A well-calibrated model exhibits a larger OOD standard deviation. (B1-B3) Mean predictions and normalized standard deviations across samples from the posterior of 3 models for IND and OOD data. Both Hybrid (B1) and VI (B2) provide appropriate uncertainty estimates on this dataset for IND and OOD. MCD fails to properly represent the uncertainty.

## 3. Results

**Toy dataset** We first run HybridBNN on synthetic data. We generate $(x_i, y_i)$ pairs using the following model.

$$w \sim \mathcal{N}(0,1),\ x_i \sim \mathcal{U}(-1,1),\ y_i|x_i, w = wx_i + \frac{1}{2}(x_i + 0.5)^2 \sin(4x_i) + \epsilon_i,\ \epsilon_i \sim \mathcal{N}(0, \sigma^2),\ (4)$$

This type of model presents a challenging calibration problem for neural nets (Blundell et al., 2015). We compared HybridBNN to VI and MCD on neural networks with the same architecture and the weight same prior. Fig. 1 shows the comparison between the models. The HybridBNN leads to higher uncertainty in regions where no data is seen, an important sign of calibration for out-of-distribution (OOD) data. We refer the reader to the Appendix A.1 for experimental details.

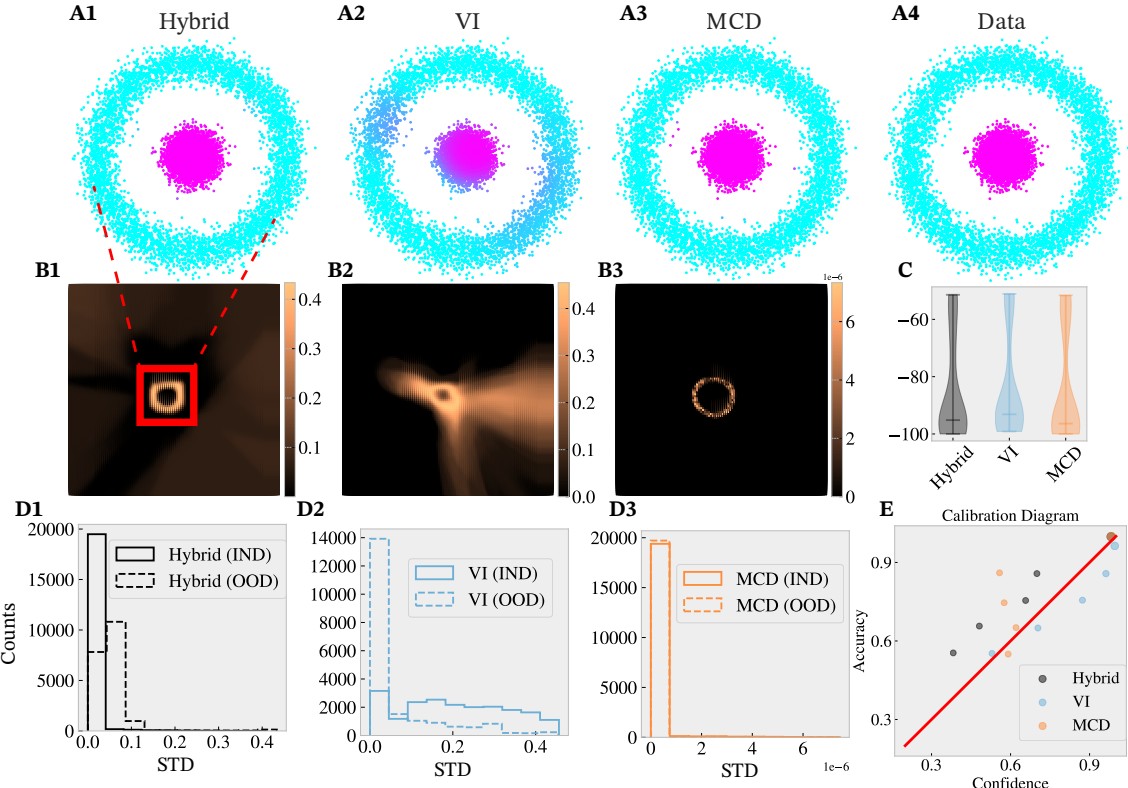

Figure 2: **Out of distribution uncertainty estimation in the two-circles dataset.** Data used for training the models to classify blue vs. pink dots is shown in panel A4. Mean prediction across posterior samples from each model is shown in panels (A1-A3). All models are able to classify data with perfect IND accuracy. (B1-B3) STD of predictions for IND (red box) and OOD data. Only HybridBNN provides a calibrated uncertainty estimate for regions outside of the training distribution. (C) IND calibrated log-likelihood across models. (D1-D3) Same as in Fig. 1A1-A3. HybridBNN is the only model that provides larger uncertainties for OOD data. (E) Calibration plot for IND, all models perform similarly for IND.

**Distance awareness**    Distance awareness is a key property of a well-calibrated statistical model that enables proper uncertainty estimation in the OOD data.    Liu et al. (2020b) discuss that models such as the Gaussian Process are distance-aware, while approximate inference algorithms for BNNs such as VI and MCD result in models that are not distance-aware. We explore this in Fig. 2 where we empirically show that HybridBNN is indeed distance aware. We further demonstrate that while all models have similar calibration performance for IND data, only HybridBNN performs well for OOD calibration. We refer the reader to the Appendix A.1 for more results and experimental details.

**Calibration metrics**    A fundamental component of machine learning models is confidence calibration or the ability to predict probability estimates representative of the true correct-

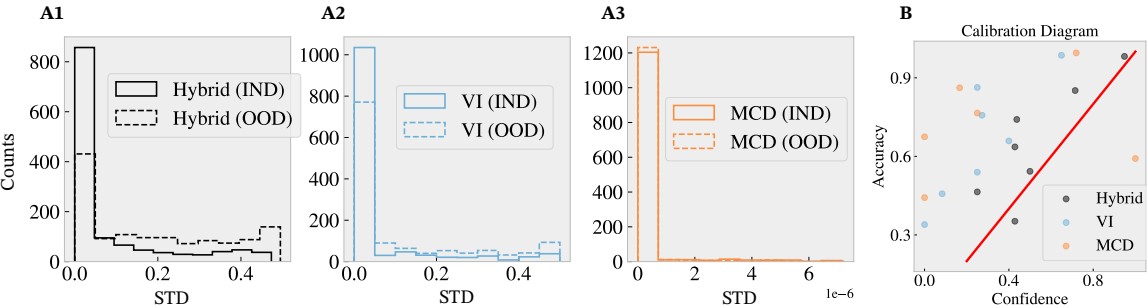

Figure 3: **Calibration evaluation on MNIST.** (A1-A3) Posterior STD for IND and OOD data. Here IND are test images from MNIST not used in training and OOD data consists of random images generated by sampling each pixel from a normal iid distribution. The Hybrid model (A1) provides larger uncertainties for OOD data. (B) Calibration plot showing that the Hybrid model has better IND calibration properties on MNIST.

ness likelihood. Guo et al. (2017a) show that modern neural networks are poorly calibrated, as they often lead to overconfident predictions (Guo et al., 2017b). Common metrics for evaluating the calibration of a model are calibration diagrams, which represent the accuracy plotted against the model's confidence, as well as the Expected and Maximum Calibration Error (ECE, MCE), defined respectively as the difference in expectation between confidence and accuracy $\mathbb{E}|\mathbb{P}(\hat{\boldsymbol{y}} = \boldsymbol{y}|\hat{p} = p) - p|$ and the maximum difference $\max_p |\mathbb{P}(\hat{\boldsymbol{y}} = \boldsymbol{y}|\hat{p} = p) - p|$ (Naeini MP, 2015). In Fig. 3B, we show the calibration diagrams for our model on the MNIST dataset. Architectural and optimization details can be found in Appendix A.1.

These diagrams and metrics are useful to evaluate our model but suffer from several problems. Estimates are biased, maximized for an infinite temperature model, and do not take into account the full probability vector but just the maximum. Ashukha et al. (2020) have benchmarked different uncertainty estimates for machine learning, pointing out many pitfalls of existing metrics. They propose another metric that does not suffer from these issues: Calibrated Log-Likelihood (CLL), which is the log-likelihood at optimal temperature (Guo et al., 2017a). We use this metric to compare the different models and estimate it following Ashukha et al. (2020) by separating the test set in a validation set to estimate the optimal temperature $T$, and a test set for evaluating log-likelihood at $T$. We then repeat this several times to get an unbiased estimate of CLL (Fig. 2C).

Finally, similarly to Lakshminarayanan et al. (2017), we evaluate the spread of the predictive distribution on OOD data and use this to evaluate the quality of the uncertainty estimates. To do this, we first train our model on the MNIST train set and evaluate the standard deviation (STD) distribution on both the MNIST test set and an OOD set created by sampling random images with iid normal pixels. Fig. 1A1-A3, Fig. 2D1-D3, and Fig. 3A1-A3 all show that our model's STD distribution has a heavier tail on OOD data, reflecting higher uncertainty.

## Conclusion

In this paper, we presented HybridBNN, a model for joint training of stochastic and deterministic weights in neural networks, and showed that it possesses desirable calibration properties on two toy datasets and MNIST. Importantly, HybridBNN possesses distance awareness, a property discussed in the literature that enables proper OOD uncertainty estimation. As opposed to previously proposed distance-aware models, HybridBNN does not introduce any architectural constraints and is compatible with most architectures (Liu et al., 2020b). An important limitation of HybridBNN is that it assumes that the epistemic uncertainty is low-dimensional. More efficient inference algorithms are needed for datasets with higher-dimensional epistemic uncertainty. In other words, HybridBNNs do not scale computationally to large Bayesian layers, which might be required if the stochasticity in the data is high-dimensional (Sharma et al., 2023). Moreover, while our implementation of HybridBNN supports stochastic weights only in the last layer, the algorithm extends beyond this setting to arbitrary stochastic weights in the architecture. In addition, while we have observed that HybridBNN produces calibrated predictions on OOD data, we do not have theoretical results supporting that. Hence, it is unclear whether these results extend to other architectures such as convolutional or recurrent neural networks. We leave these open questions for future work.

## Acknowledgments

The authors express their gratitude to Geoff Pleiss and John Cunningham for their helpful comments and discussions throughout the development of the ideas presented in the paper.

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

# Appendix A. Appendix

## A.1. Experimental Details

**Regression Model** We generated 128 pairs $(\boldsymbol{x}, \boldsymbol{y})$ for training and a batch size of 32. We set the value of $\sigma^2$ to 0.2. We trained a HybridBNN with 2 deterministic layers of size 10 and a stochastic layer of size 10. The `tanh` activation was used for all layers except for the last layer. For updating the deterministic weights, we used `Adam` optimizer with a learning rate of $10^{-3}$. The same architecture was used for the VI and MCD models. In all models, we placed a $\mathcal{N}(0,1)$ prior over the weights of the networks. Convergence was assessed by inspecting the loss value and ensuring that it does not change more than $10^{-4}$ for at least 10 iterations. The learning rate for other models was set to ensure that the networks do not diverge. For VI we used a value of $10^{-2}$ and for MCD we used $10^{-1}$. The training data was generated by sampling $\boldsymbol{x}$ uniformly on the interval $(-1, 1)$ and computing $y$ according to Eqn 4. The OOD data was generated by sampling $x$ uniformly on the interval $(1, 3)$. For HybridBNN, we inferred a posterior over the No U-Turn Sampler (NUTS) with 1 chain, 1000 burn-in samples, and 100 posterior samples. We used a normal likelihood for the predictions which translated into MSE loss for updating deterministic weights.

**Two Circles Classification** In this experiment, we used a training set of size 128 sampled independently from `make_circles` function in `sklearn` with a noise value of 0.1. We then generated the 2D $\boldsymbol{x}$ vectors into 10-dimensional space using an orthogonal matrix to facilitate training. For HybridBNN architecture, we used 2 deterministic layers of size 200 and a stochastic layer of size 10. We used `relu` activation for all but the last layer. The last layer used a `softmax` activation for the classification. The same architecture was used for VI and MCD. In this experiment, we generated 500 samples from the posterior in each iteration of posterior inference. The likelihood model used for the classification was categorical which translated into cross-entropy loss for updating the deterministic weights. The rest of the parameters are similar to the regression experiment. To show the convergence of the model, Fig. 4 shows that the model not only provides better predictive performance through iterations, but it improves distance awareness and OOD calibration too.



Figure 4: **HybridBNN improves OOD calibration through the iterations of the algorithm.**

**MNIST** We uniformly sampled 128 images from different digit classes in the MNIST dataset and reformatted the data to a 784-dimensional vector. For the HybridBNN architecture, we used two deterministic layers of size 200 and a stochastic layer of size 20. The `relu` activation was used for all but the last layer. For the last layer, we used `softmax` to

match the classification task. We used a learning rate of $10^{-1}$ for updating the deterministic weights of HybridBNN. The same value was used for the VI model. The learning rate for the MCD model was chosen to be $10^{-2}$. Other parameters are similar to the two circles experiment.

