# OpenReview forum: "HybridBNN: Joint Training of Deterministic and Stochastic Layers in Bayesian Neural Nets"
_approximateinference.org/AABI/2024/Symposium — AABI 2024_

### Official Review · Reviewer_xjAV · 2024-04-22
**Simple yet effective idea with only the most basic evaluation**

**Rating:** 6
**Confidence:** 3

**Review:**

The submitted paper proposes to train BNNs by dividing the networks weights into deterministic and stochastic parameters and training them via expectation-maximization. If there are few stochastic parameters, such a training approach can be more effective than some alternatives while still providing high-quality uncertainty representation (as demonstrated in experiments).

The proposed approach is clearly presented and easy to follow. The basic idea of constructing hybrid BNNs is not novel (as also mentioned in the paper) but the specific way of training it seems to no have been considered before. In experiments, improved performance of the proposed approach over Monte Carlo Dropout and Variational Inference based approaches on toy data and MNIST are demonstrated in terms of the uncertainty calibration.

While there is some evidence that the proposed approach can be useful, the empirical evaluation is quite limited. Also very basic comparisons, e.g., with the referred paper that proposed the division of the papers and builds on Bayes-by-Backprop, is not included. Furthermore, scalability of the proposed approach to more relevant real-world settings is not demonstrated.

Minor:
* Please update the reference style/how you refer articles. In the current form, the reading flow is often interrupted.
* There is sufficient space in the paper, you can add more details and a pseudo code for your approach.

---

### Official Review · Reviewer_NY9x · 2024-04-24
**impossible to review**

**Rating:** 2
**Confidence:** 4

**Review:**

This paper employs an expectation maximisation algorithm to a Bayesian neural network with both deterministic and stochastic parameters. In principle, this might be a good idea. Unfortunately, the paper is so poorly written that I find it impossible to review.
The equations are completely messed up. Eq. (2) and above: I suppose you integrate over theta_s (the volume element is missing in some of the integrals). Are you using the log of the integral or the integral of the log? This is not consistent in your equations. Then, in the E-step, you take the expectation w.r.t. a distribution of theta_s. However, the quantity of which you take this expectation value does not depend on theta_s because it has been integrated out. Please derive your theory properly and do not leave it to the reader to guess what you are doing.
Also the examples are not properly explained: In eq (4), I suppose W is a matrix? What are its properties and its dimensions? Example 2 is not explained at all. Please also explain what you mean by distance awareness. I´m happy to review a resubmission of this work once it is properly done.

---

### Official Review · Reviewer_o6hH · 2024-04-24
**Interesting approach, scalability might be an issue but at this stage the paper can be improved**

**Rating:** 5
**Confidence:** 3

**Review:**

The paper deals with subspace approximate inference for Bayesian neural networks where some weights are treated as deterministic while others are treated as stochastic. They propose to treat the stochastic weights as latent variables and propose an expectation-maximization procedure for training jointly both deterministic and stochastic weights. This EM procedure involves getting samples from the posterior at each training step.

**Strength**

- The EM procedure to jointly learn the stochastic and deterministic weights seems novel.
- Paper is well and clearly written and easy to read.

**Things that can improve the paper**

There are several aspects of the paper that are not fully convincing and I think that addressing these points will really make the paper stronger.
- Fig. 1: I am assuming that the grey points are the training samples, but it seems that all the mean functions are off or biased, while you write in the caption that all models perform well on mean prediction.
- Fig. 1/Regression experiment: Since the network is really small it would be possible to get samples from the true posterior over all the weights of the network by running HMC, it would be interesting to compare the behavior with such an approach.
- I feel that a discussion on how to choose which weights to treat as deterministic and which to consider as stochastic is missing. Do you get the same behaviors that [1] also experienced?
- I also have the feeling that scaling this method to big networks might be expensive. Indeed at every step, you have to get samples from the posterior of a subset of the weights. It seems that in that case, we will have to rely still on approximation to get some samples, which is also the same problem we are interested in solving in the first place.
- MNIST example: if I am understanding correctly you are treating only the last layer weights as stochastic. It seems that the procedure you are presenting is way more expensive compared to last-layer Laplace, for example. I know that last-layer Laplace is a two step procedure, while your proposed method is doing everything jointly. But it would have been interesting to compare the two and discuss which are the advantages of performing a joint optimization over using post-hoc Laplace.


**References**

[1] Sharma, Mrinank, et al. "Do Bayesian Neural Networks Need To Be Fully Stochastic?." International Conference on Artificial Intelligence and Statistics. PMLR, 2023.

---

### Official Review · Reviewer_aGFx · 2024-04-24

**Rating:** 5
**Confidence:** 5

**Review:**

The Authors propose a method (HybridBNN) to perform approximate inference in Bayesian neural networks by using a mixture of deterministic and stochastic weights.

Comments:

- The idea of having deterministic and stochastic weights in the model is not new but the authors provide a fair list of references of related works (even though it misses some important ones, for example on last-layer Laplace approximations [e.g., 1]).
- The idea of framing this problem as an EM algorithm is interesting, as it allows joint optimization of both sets of weights.
- Besides some common issues of hybrid models (such as, how much of the weights should be deterministic and how much should be stochastic), it's not completely clear to me how the convergence behaves in practice, and how the method scales to larger models.
- Claming "state-of-the-art prediction and calibration performance" (from the abstract) after only comparing with toy datasets and without other competitive methods (at least, HMC on the full model) is not very convincing (and I would encourage to amend this comment).

[1] Kristiadi et al. Being Bayesian, Even Just a Bit, Fixes Overconfidence in ReLU Networks.

---

### Official Review · Reviewer_2Jtd · 2024-04-24
**Two and a half drawbacks of the HybridBNN Paper**

**Rating:** 6
**Confidence:** 3

**Review:**

The paper on HybridBNNs presents a simplified way to use subspaces for the construction of Bayesian Neural Networks (BNNs). More concretely, they split the parameter space of a Neural Network (NN) into two sets of either deterministic or stochastic weights. The training procedure then follows an Expectation Maximisation (EM) principle, where an average output (Expectation) is constructed using fixed deterministic and sampled stochastic weights (Monte Carlo Integration). The deterministic weights are then updated (Maximisation) using conventional back-propagation through the derived expectation value (as a function of the deterministic weights). The method is shown to work on two low-dimensional toy models for regression and classification, respectively, as well as for MNIST. The results are contrasted with ones obtained using, instead of HybridBNN, Variational Inference (VI) or Monte Carlo Dropout (MCD) as well established methods for uncertainty quantification.The results are mostly reasonably explained and highlight key properties of the method. Of interest, as claimed by the authors, are calibration and prediction performance, "distance awareness", scalability to "high-dimensional inputs" and being less "mathematically involved".
The presented method fits well into the scope of AABI as it provides a simple approach to implement approximations of BNNs and the evaluation gives convincing evidence that it works competitively on the examples provided. However, there are three (larger) drawbacks concerning the work.
The first regards the related work "section" (middle of the introduction), which, while well written, lists only two works later than 2020 (neither of which provide an approach to uncertainty quantification, at least if judged by their title). This makes it harder to proper contextualize the work and omits recent advances, e.g. "Effective Bayesian Heteroscedastic Regression with Deep Neural Networks" by A. Immer et al (NeurIPS2023).
The second point is the evaluation of the method. On the one hand the results of HybridBNN are convincing, on the other the advantages compared to other methods are not fully apparent (or valid?). W.r.t. the latter, the evaluation of MCD looks wrong. Specifically, the standard deviations reported in figures 1 to 3 are on scales of $10^{-6}$ suggesting that the dropout rate might be wrongly calibrated and the used data is more numerical noise than meaning (note for others: the visualisations in the figures, eg panel B3 in Fig1, use re-scaling). As I did not find technical details on the implementations of VI, MCD in the appendix (beyond what is shared with the HybridBNN) it is difficult to judge this further. Regarding the advantages, "distance awareness" seems to have no (mathematical) definition. Assuming it implies a growth of epistemic uncertainty with distance from the in-data distribution (IND) figure 1 shows that HybridBNNs have this property, but VI (to a slightly lesser degree) possesses them as well. With the limited amount of evaluation (3 datasets) it is therefore hard to be certain on the improvement (e.g. also MCD is known to have examples showcasing distance awareness, in other works, compare "Wasserstein Dropout" by J. Sicking et al (Springer Machine Learning, 2022)). Based on the appendix, it also appears that the evaluation on "real" data (MNIST) uses only 128 samples (potentially per class) out of 70,000 available data points. Why?
Third, I'm not fully confident I could replicate the shown results based on the information provided. Specifically, the update of the stochastic weights seems unclear. In eq. (2) the Expectation (E) is taken over the conditional $p(\theta_s|\hat\theta_d, x,y)=:p_s$, which indicates a dependence on the training samples $(x,y)$. In fact, it is mentioned (also in sec. A.1) that an MCMC approach is used for sampling. However, the distribution $p_s$ is not given or updated (as far as i understand, except for changes in $\theta_d$), should be unavailable at runtime (i.e. during inference), and seems to play no role in the Monte Carlo approximation of this expectation value, compare eq. (3). Maybe this point could be clarified more, e.g., in the appendix. It might also have implications for run-time, which seems to be not discussed in the paper. Further, based on the architecture descriptions in A.1 i was left with the (potentially wrong) impression that the stochastic layers are the last layers, as they are always mentioned last in the description. Maybe this can also be clarified ,e.g., in context of the generality of architecture that is aimed for (if true, it reminded of "Last Layer MCD" variants).

In summary, the paper demonstrates an, in principle, working and interesting method, that produces reasonable and plausible results and fits well into the scope of the workshop. But based on the drawbacks detailed above it is, sadly, not possible to rate the work better than "marginally above acceptance threshold". (None of the drawbacks invalidate the method itself. But especially the first two seem to be severe in light of good scientific practice, giving me a torn feeling)

---

### Official Review · Reviewer_oead · 2024-04-26
**Reject: HybridBNN**

**Rating:** 4
**Confidence:** 3

**Review:**

## Summary
The extend abstract proposes HybridBNN, a method for training Bayesian Neural Networks (BNNs). HybridBNN relies on partitioning the weights of the neural network into two classes: deterministic weights and stochastic weights. After partitioning, HybridBNN treats the deterministic weights as parameters of a latent variable model where the stochastic weights are the latent variables. This view allows the application of the expectation-maximization (EM) algorithm to train the network. The authors present experimental results on applying HybridBNN to the MNIST dataset, and two "toy datasets". The supplementary material provides the experimental details.

## Discussion
The following points suggest that a more refined version of the abstract provides more value to the AABI community and generate better feedback for the authors.

1. The second equation from the top on page three is incorrect. The order of integration and taking the logarithm are reversed; i.e. we should have logs of integrals, not integrals of logs. Moreover, the variables of integration are not indicated; i.e. a $d\theta_s$ is missing from both integrals.

2. Equation (2) is also incorrect. The E-step of the EM algorithm calculates the expectation of the joint density, not the marginal density. That is, the expectation of $p(x)p(\theta_s)p(y|f_\theta(x))$ should be calculated, not its integral. The text of the extended abstract also incorrectly refers to the log marginal.

3. The last equation in section 2 contains a typo: the summation index is $i$ while $x$ and $y$ are indexed by $j$.

4. The details of the MCMC algorithm used for the second and third algorithms is not discussed. The number of iterations of the EM algorithm is also not discussed. This makes the experimental results hard to reproduce.

5. The bias of the trained BNNs is never reported. All three experiments show plots that characterize the trained predictors as low-variance for the in-distribution data and higher-variance for out-of-distribution data. Yet, no indication is given that the bias is low for in-distribution data.

6. No calibration plots are given for the first experiment. It is merely reported that the HybridBNN network reports higher uncertainty for out-of-distribution data, but this higher uncertainty might well be uncalibrated.

7. The calibration plot in Figure 2 actually shows that the HybridBNN is under-confident and some of the VI and MCD networks are better calibrated.


The idea behind the extended abstract is simple and practical. The EM algorithm is a popular method for maximum likelihood estimation in the machine learning literature, and its application to BNNs is sensible. I am not familiar with the literature on Bayesian neural networks, and so it surprises me that the EM algorithm has not been applied to the training of BNNs. As such, HybridBNN has the potential to become a widely used method for the training of BNNs. I look forward to a future submission addressing the above points.

---

### Official Review · Reviewer_FN4W · 2024-04-26
**A timely contribution to the field of hybrid BNNs, yet the scope of its applications is disappointingly narrow**

**Rating:** 6
**Confidence:** 5

**Review:**

# Summary Of Contributions
The work introduces HybridBNN, a new subspace inference method for Bayesian Neural Networks (BNNs) that separates network weights into deterministic and stochastic subsets to address the challenges of high dimensionality and intractable exact inference in traditional BNNs. HybridBNN, using an expectation-maximization algorithm for joint inference and optimization, demonstrates superior prediction and calibration performance across various datasets, offering a promising approach for developing generally applicable, calibrated models.

# Strengths And Weaknesses
### Strengths
- Bayesian deep learning represents a promising avenue, and efforts to scale Bayesian Neural Networks are both timely and essential.
- Overall, the presentation of the material is well-executed.

### Weaknesses
- The main text lacks clarity regarding the division of weights into deterministic and stochastic subsets.
- The appendix reveals that all examples in the paper employ only the last layer as stochastic, which is disappointing for two reasons:  1. this concept of last-layer Bayesian Neural Networks is not new, 2. it represents a limited approach.  Could the authors explore alternative configurations for these subsets?

# Requested Changes
- Equations are presented in an unconventional manner; many are not directly related to the preceding text, as seen with all equations on page 3. Please integrate these equations into the text more fluidly, ensuring appropriate punctuation follows each equation.
- Similarly, citations should be incorporated parenthetically unless they are an integral part of the sentence, as noted at the top of page 2. Use \citep{} for such instances.
- The differential \(d\theta_s\) is missing in the integrals on page 3, which is critical for clarity and mathematical accuracy.
- Regarding the Appendix's methodology of using 1000 burn-in cycles for only 100 retained posterior samples, this approach is unconventional given the complexity of the models. Consider employing multiple parallel chains and discuss the convergence of these chains, potentially using the R-hat statistic for a more robust analysis.
- In the MNIST section of the Appendix, replace 'the date' with 'the data'. Additionally, include a discussion on the convergence of the chains to validate the robustness of your findings.

---

### Meta-Review · Area_Chair_pV7F · 2024-05-12

**Recommendation:** Accept (Poster)
**Confidence:** 4

**Metareview:**

This paper proposes a way to train a subnetwork BNN. First, the network's parameters are partitioned into stochastic and deterministic parts. Then the former is treated as a latent variable and the latter a hyperparameter. Then, naturally, the marginal-likelihood maximization can be applied. The authors proposed to maximize it via an EM scheme.

This paper is quite borderline. While some reviewers think that the paper is quite interesting, the majority agree that some more work need to be done to make this paper more well-written. First, the method exposition itself (eqs. 2 and 3) is unclear, making some reviewers doubt its validity. Second, the authors overclaimed their results---they claimed SotA results while only shows experiments on toy and MNIST datasets. Lastly, the authors only study the last-layer stochastic weights; however they did not compare against standard last-layer BNN baselines like the last-layer Laplace. The latter can also be trained via an EM scheme using the Laplace marginal likelihood (see Immer et al., ICML 2021).

I tend to give the paper the benefit of the doubt (slightly). But I wholeheartedly agree with the reviewers that the paper must be improved for the final version.

---

### Decision · Program_Chairs · 2024-05-27

Accept